# Effects of 12-week Aerobic Exercise on Arterial Stiffness, Inflammation, and Cardiorespiratory Fitness in Women with Systemic LUPUS Erythematosus: Non-Randomized Controlled Trial

**DOI:** 10.3390/jcm7120477

**Published:** 2018-11-24

**Authors:** Alberto Soriano-Maldonado, Pablo Morillas-de-Laguno, José Mario Sabio, Blanca Gavilán-Carrera, Antonio Rosales-Castillo, Cristina Montalbán-Méndez, Luis Manuel Sáez-Urán, José Luis Callejas-Rubio, José Antonio Vargas-Hitos

**Affiliations:** 1Department of Education, Faculty of Education Sciences, University of Almería, 04120 Almería, Spain; 2SPORT Research Group (CTS-1024), CERNEP Research Center, University of Almería, 04120 Almería, Spain; 3Department of Physical Education and Sport, Faculty of Sport Sciences, University of Granada, 18071 Granada, Spain; pmorillasdelaguno@hotmail.com (P.M.-d.-L.); bgaviera@gmail.com (B.G.-C.); 4Systemic Autoimmune Diseases Unit, Department of Internal Medicine, “Virgen de las Nieves” University Hospital, 18014 Granada, Spain; jomasabio@gmail.com (J.M.S.); anrocas90@hotmail.com (A.R.-C.); lm_889@hotmail.com (L.M.S.-U.); joseantoniovh@hotmail.com (J.A.V.-H.); 5Service of Endocrinology, La Mancha Centro Hospital, 13071 Ciudad Real, Spain; cmontalbanm@sescam.jcam.es; 6Systemic Autoimmune Diseases Unit, Department of Internal Medicine, “San Cecilio” University Hospital, 18012 Granada, Spain; jlcalleja@telefonica.net

**Keywords:** exercise, physical fitness, aerobic fitness, physical function, pulse wave velocity, autoimmune diseases, vascular health, atherosclerosis, lupus, inflammatory disease

## Abstract

This study assessed the effect of 12-week aerobic exercise on arterial stiffness (primary outcome), inflammation, oxidative stress, and cardiorespiratory fitness (secondary outcomes) in women with systemic lupus erythematosus (SLE). In a non-randomized clinical trial, 58 women with SLE were assigned to either aerobic exercise (*n* = 26) or usual care (*n* = 32). The intervention comprised 12 weeks of aerobic exercise (2 sessions × 75 min/week) between 40–75% of the individual’s heart rate reserve. At baseline and at week 12, arterial stiffness was assessed through pulse wave velocity (PWV), inflammatory (i.e., high-sensitivity C-reactive protein [hsCRP], tumor necrosis factor alpha [TFN-α], and inteleukin 6 [IL-6]) and oxidative stress (i.e., myeloperoxidase [MPO]) markers were obtained from blood samples, and cardiorespiratory fitness was assessed (Bruce test). There were no between-group differences in the changes in arterial stiffness (median PWV difference −0.034, 95% CI −0.42 to 0.36 m/s; *p* = 0.860) or hsCRP, TNF-α, IL-6, and MPO (all *p* > 0.05) at week 12. In comparison to the control group, the exercise group significantly increased cardiorespiratory fitness (median difference 2.26 minutes, 95% CI 0.98 to 3.55; *p* = 0.001). These results suggest that 12 weeks of progressive treadmill aerobic exercise increases cardiorespiratory fitness without exacerbating arterial stiffness, inflammation, or oxidative stress in women with SLE.

## 1. Introduction

Systemic lupus erythematosus (SLE) is an autoimmune disease of an unknown etiology that predominantly affects young adult women [1], with a prevalence of up to 1 case in 1000 inhabitants in Europe [2,3]. Its prognosis has significantly improved in recent decades leading to new comorbidities, such as atherosclerotic cardiovascular diseases (CVD) [4], which have become a major cause of mortality in this population [5]. It has been shown that classical CVD risk factors (e.g., hypertension, dyslipidemia, diabetes, smoking etc.) fail to fully describe the excess of CV morbi-mortality observed in SLE [4,6], and novel CV risk factors, such as arterial stiffness, systemic inflammation, or oxidative stress, are involved [4,7,8].

Arterial stiffness is a marker of subclinical atherosclerosis that increases with age and is significantly elevated in patients with SLE compared to the general population [8,9]. Arterial stiffness reveals structural changes in the elasticity of the arteries prior to the development of clinical atherosclerosis, and is a powerful predictor of CVD independently of other CV risk factors, both in the general population [10] and in patients with SLE [11]. Therefore, attenuating the increase of arterial stiffness in this population is of clinical relevance for the early prevention of CVD.

In SLE, inflammation and tissue damage are mediated by pro-inflammatory cytokines, such as tumor necrosis factor alpha (TNF-α) and interleukin-6 (IL-6), released by recruited inflammatory cells (macrophages, myeloid dendritic cells, pathogenetic T and B cells) and immune complexes-induced complement activation [12]. These markers of systemic inflammation, as well as C-reactive protein (CRP), are known to independently predict CV events [13,14], and represent potential mechanisms explaining the excess CV morbi-mortality in SLE [15]. Oxidative stress (i.e., an imbalance in the cells due to an increase in free radicals or a decrease in antioxidants) has also shown to be involved in the development of CVD in patients with SLE [7,16]. Although behavioral interventions, such as regular exercise, might elicit significant CV benefits without many of the side effects of pharmacological treatments, they are understudied and hardly appear in the British Society for Rheumatology guidelines for the management of SLE [17].

Exercise has been shown to increase physical fitness and reduce CVD risk in many populations across the whole lifespan. The American College of Sports Medicine (ACSM) highlights the need to undertake a minimum of 150 min/week (i.e., accumulated in bouts of ≥10 min) of aerobic exercise of moderate to vigorous intensity in adults [18]. In a sample of women with SLE with mild/inactive disease, we cross-sectionally observed no association between accelerometer-assessed physical activity and arterial stiffness [19], although a higher level of cardiorespiratory fitness was related to lower age-related arterial stiffness in this population [20]. Although aerobic exercise has a promising role attenuating arterial stiffness in the general population [21], its effects in women with SLE have not been previously investigated.

Perandini et al. observed that a single bout (i.e., 30 min) of aerobic exercise (i.e., either at 50% or 70% of VO_2peak_) did not increase inflammation in women with either active or inactive SLE [22], and that 12 weeks of aerobic training (*n* = 8) tended to reduce inflammation in comparison to a control group (*n* = 10) [23]. Timoteo et al. [24], however, did not observe changes in IL-6 or TNF-α in the combined exercise (i.e., flexibility, resistance and aerobic training) group (*n* = 5) compared to the control group (*n* = 9). Overall, these studies had a small sample size and the exercise programs were not comprehensively described to allow replication or application in clinical practice.

The primary aim of this study was to assess the effect of a 12-week aerobic exercise intervention following the ACSM guidelines on arterial stiffness in women with SLE in comparison with usual care. Secondary aims were to assess the effects of the exercise intervention on inflammation, oxidative stress, and cardiorespiratory fitness. We hypothesized that the intervention would have a positive impact on CV risk factors and cardiorespiratory fitness [25] in women with SLE.

## 2. Methods

### 2.1. Design and Protocol Registration

This non-randomized controlled trial was registered at clinicaltrials.gov [NCT03107442] on 11 April 2017, before the enrolment of participants started (i.e., on 12 April), and no deviations occurred regarding the primary outcome and the secondary outcomes analyzed here.

### 2.2. Setting and Eligibility Criteria

Participants were recruited from the Systemic Autoimmune Diseases Unit of the “Virgen de las Nieves” and the “San Cecilio” University Hospitals. Women with a diagnosis of SLE according to the ACR criteria [26], a follow-up of ≥12 months, clinical and treatment stability during the previous six months, and not performing regular exercise (defined as ≥60 min/week of structured exercise) were included. Exclusion criteria were to have been under biological treatment in the previous six months or to need a prednisone dose of >10 mg/day; a background of CVD in the previous year; to present contraindications to perform exercise; other associated rheumatic conditions; pregnancy; active acute or chronic infection; neoplasms; acute renal failure; cardiac or pulmonary involvement; body mass index (BMI) >35; or not being able to read, understand, and sign written informed consent. All participants received detailed information about the study procedures, and signed written informed consent. The Research Ethics Committee of Granada approved the protocol on 11 November 2016 (reference No.: 10/2016).

### 2.3. Procedures

A telephone screening was conducted. Potentially eligible participants were invited to a personal screening and, if included, day 1 of the baseline examination was performed. The baseline examination comprised two assessment days. On day 1, pulse wave velocity (PWV) was assessed. Thereafter, cardiorespiratory fitness testing was performed, and socio-demographic and clinical information was collected. On day 2 (i.e., between two and four days after day 1), 8-h fasting blood samples were collected between 8:00 a.m. and 10:00 a.m. This article follows the TREND statement for improving the reporting of non-randomized experiments of behavioral and public health interventions (downloadable at EQUATOR Network: https://goo.gl/ZSyLrj; Appendix A) [27]. The funding source had no role in the study.

### 2.4. Interventions

#### 2.4.1. Exercise Group

To maximize transparency and replicability, the exercise program described in this manuscript follows the Consensus on Exercise Reporting Template (CERT; Appendix A) [28]. The patients assigned to exercise performed two 75-min sessions per week during a total of 12 weeks (i.e., 24 sessions) of moderate to vigorous intensity aerobic exercise on a treadmill (BH, Serie i.RC12 Dual, Vitoria-Gasteiz, Spain) from 24 April to 14 July 2017. A scheme of the prescribed exercise intervention is presented in Table 1.

The sessions took place in a quiet room of the “Virgen de las Nieves” Hospital, Granada (Spain). All sessions were performed in groups of a maximum of five persons (depending on the patients’ schedule preferences) and were supervised by both exercise professionals with a degree in Sports Sciences and residents from the Internal Medicine Department. Attendance at the sessions was registered daily and patients were contacted upon any missing session to ask for the reason and motivate them to replace it on an alternative day of the same week. Adherence to exercise is reported as the median attendance frequency and the proportion of patients attending ≥75% (i.e., 18 sessions; the minimum pre-defined attendance to assess efficacy) and ≥90% of the sessions. All the sessions began with a warm-up comprising 3–4 min of activation on the treadmill at about 35–40% of the heart rate reserve (HRR) and 3–4 min of active stretching of major muscle groups, and ended with a cool down phase of static stretching of major muscle groups and relaxation. Exercise was individually prescribed to represent moderate-to-vigorous intensity, with training intensity ranging from 40% to 75% of each patient’s HRR. The maximum heart rate (HR_max_) was estimated with the formula by Tanaka et al. (HR_max_ = 208 − (0.7 × age)) [29]. The training (or target) heart rate (tHR) was calculated with the formula tHR = HR_rest_ + (%HRR). Heart rate was continuously monitored during all sessions (Polar V800, Kempele, Finland). We used the session rating of perceived exertion (RPE) as a measure of subjective training load [30], and the feeling scale to assess positive affective responses experienced before and after each session [31].

The starting level was specific for each individual according to her previous exercise experience and physical fitness. During the first half of the program, only continuous exercise was performed so that the patients got used to the treadmill and felt confident at increasing intensities. Continuous sessions comprised several bouts of exertion at constant intensity, followed by a couple of minutes of recovery (i.e., rest) to drink water. During month 2, there were alternated continuous and interval sessions, and at month 3, the patients undertook interval training sessions, where there were periods of lower and periods of higher intensity efforts followed by some minutes of rest for hydration (Table 1). The progression in volume and/or intensity was patient-limited and was undertaken by increasing the treadmill speed (first) or inclination according to the symptoms and perceived exertion. There were no home-based or non-exercise components within this intervention. However, if a participant was eventually not able to attend a particular session, we provided her with a heart rate monitor and allowed recovery of that session out of the Hospital (a total of seven sessions were recovered in this fashion). Finally, the exercise intensity progressions had to be slightly modified from the initial plan. For instance, several patients perceived a 5% HRR intensity increase (i.e., from one week to another) as very heavy and difficult-to-follow. Consequently, there were weeks in which exercise intensity increased by 2.5% instead of 5% (Table 1).

#### 2.4.2. Control Group

After the baseline evaluation, the SLE patients assigned to the (usual care) control group received verbal information about a healthy lifestyle, including physical activity guidelines and basic nutritional information.

### 2.5. Outcome Measures

#### 2.5.1. Primary Outcome Measure: Arterial Stiffness

Arterial stiffness was assessed in a sitting position by PWV [9], using the Mobil-O-Graph^®^ 24 h pulse wave analysis monitor (IEM GmbH, Stolberg, Germany), whose operation is based on oscillometry recorded by a blood pressure cuff placed on the brachial artery. The coefficient of variation (CV) of Mobil-O-Graph for consecutive PWV analyses is 3.4% and its intraclass correlation coefficient is 0.98 [0.96–0.99] [32]. This device has been largely shown to be valid and reliable for measuring PWV and central blood pressure in different populations [33,34], meets the accuracy requirements of the British Hypertension Society (BHS) standard [35], and can be recommended for clinical use [36].

#### 2.5.2. Secondary Outcome Measures

##### Blood Samples and Biochemical Analyses

Fasting blood specimens for biochemical and immunological tests were collected and routinely processed by the central laboratory of our hospital. Among other measurements, they included lipids, insulin (BioRad, Marne-la-Coquette, France), and a routine biochemical profile. The homeostatic model assessment for insulin resistance (HOMA-IR) was calculated (HOMA-IR = glucose (mmol/L) × insulin (µU/L)/22.5).

##### Inflammatory Markers

Serum high-sensitivity CRP was assessed by an immunoturbidimetric method using the ARCHITECT cSystems (MULTIGENT CRP Vario assay); the limit of quantitation was 0.2 mg/L and the upper limit for normal serum was 5 mg/L (coefficient of variation <6%). Interleukin 6 and TNF-α, as well as myeloperoxidase (MPO; as marker of oxidative stress), were measured in plasma. Serum was initially separated by centrifugation and stored at −70 °C. Bioserum concentrations of IL-6/TNF-α (pg/mL) and MPO (ng/mL) were measured by an immunoradiometric assay using commercial kits (MILLIPLEX MAP Kit Human High Sensitivity T Cell Magnetic Bead Panel (HSTMAG-28SK) and Human Cardiovascular Disease Magnetic Bead Panel 2 (HCVD2MAG-67K)), Millipore) following the manufacturer’s instructions. Quantitative data were obtained by using the Luminex-200 system (Luminex Corporation, Austin, TX, USA), and data analysis was performed on XPonent 3.1 software (Austin, TX, USA). The detections limits were 0.73 pg/mL for IL-6, 0.43 pg/mL for TNF-a, and 0.024 ng/mL for MPO.

##### Cardiorespiratory Fitness

Cardiorespiratory fitness was assessed with the Bruce submaximal treadmill protocol [37]. The test comprised five increasing workload stages of 3 min each (stage 1: 2.7 km/h and 10% inclination; stage 2: 4 km/h and 12% inclination; stage 3: 5.5 km/h and 14% inclination; stage 4: 6.8 km/h and 16% inclination; stage 5: 8 km/h and 18% inclination). The test concluded when the participant achieved 85% of the individual’s HR_max_, as estimated with the formula by Tanaka et al. [29]. As validated SLE-specific formulas to estimate VO_2max_ are not available, we used the total time to reach 85% HR_max_ as the outcome of interest.

#### 2.5.3. Other Measurements

All participants filled out a socio-demographic and clinical data questionnaire. Height (cm) was measured using a height gauge, weight (kg) with a bioimpedance device (InBody R20, Biospace, Seoul, Korea), and body mass index (BMI) was calculated (kg/m^2^). Blood pressure was measured with Mobil-O-Graph^®^ (IEM GmbH, Stolberg, Germany) [36]. Disease activity was assessed through the Systemic Lupus Erythematosus Disease Activity Index (SLEDAI, range 0–105 where a higher score indicates higher degree of disease activity). Physical activity was self-reported at baseline and at week 12 with the International Physical Activity Questionnaire [38].

### 2.6. Sample Size

The sample size was calculated for the primary outcome (i.e., PWV). Ashor et al. found an average effect of aerobic exercise on PWV of −0.63 m/s in adults aged ≥18 years [21]. A total of 52 patients (26 per group) were needed to detect an effect of −0.63 (SD 0.75) m/s, with a power of 85% and an α error of 0.05. Anticipating a maximum loss to follow-up of 15%, we aimed at recruiting a total of 60 patients.

### 2.7. Treatment Allocation and Blinding

Randomization was not feasible because more than half of the patients who regularly attend the Autoimmune Disease Units lived far from the Hospital and were not able to attend twice per week in case of being randomized to exercise. Therefore, participants from the city of Granada were included in the exercise group and participants living outside Granada were included in the control group. To minimize potential selection bias, we aimed to match the groups by age (±2 years), BMI (±1 kg/m^2^), and SLEDAI (±1 unit). The data analyzer was blinded to the patient allocation.

### 2.8. Statistical Analysis

The distribution of the main study variables was assessed through histogram and Q-Q plots. As the main outcomes were non-normally distributed, their descriptive characteristics were presented using the median and interquartile range instead of the mean and standard deviation and we used non-parametric tests for the main analyses. Between-group baseline characteristics were compared with the Student *t*-test (when normally distributed) or Kruskal-Wallis test (when non-normally distributed) for continuous variables and the Chi-square test for categorical variables. The between-group differences in the studied outcomes were assessed through quantile regression with baseline values, resting heart rate (bpm) [39], and changes in physical activity (min/week; since the change in self-reported physical activity at week 12 was >60 min higher in the control compared to the exercise group and this change was associated with changes in PWV; r_Pearson_ = −0.27, *p* = 0.048) as potential confounders, after checking baseline group comparisons. As we aimed at assessing efficacy, the primary analyses were defined as per-protocol, where patients from the exercise group were included if attendance at the exercise sessions was ≥75%. To assess the robustness of the results, subsequent sensitivity analyses (i.e., baseline observation carried forward (BOCF) imputation; per-protocol with minimum attendance of ≥90%, and complete-case analyses) were conducted. A blinded investigator (AS-M) undertook the data handling and all hypothesis testing. All the analyses were conducted with Stata v.13.1 (StataCorp LP., College Station, TX, USA). Statistical significance was set at *p* < 0.05.

## 3. Results

A total of 190 women with SLE were invited to participate. The flowchart of the study participants throughout the trial is presented in Figure 1. A total of 58 patients volunteered to participate, met the inclusion criteria, signed informed consent, and were finally included and assigned to either the exercise group (*n* = 26) or the control group (*n* = 32). The median attendance at the exercise intervention was 22.5 (out of 24; i.e., ~94%) sessions. A total of 22 participants (~85%) attended ≥75% of the sessions (i.e., and were included in primary analyses) and 18 (~69%) attended ≥90% of the sessions. One participant withdrew at week 5 due to severe sciatica (not associated with the exercise program). A total of four participants (i.e., 6.8% of the total sample) were lost to follow-up in the control group at week 12 and none in the exercise group. A summary of the average exercise intensity (i.e., HR) achieved at each session, RPE, and pre- and post-session affective responses are presented in Appendix A. There were no adverse events occurring during the exercise sessions.

At baseline (Table 2), the control group showed a lower resting heart rate (mean difference −9.6 bpm; *p* = 0.002), and higher IL-6 levels (median difference 3.09 pg/mL; *p* = 0.026) than the exercise group. There were no other significant between-group differences at baseline (all *p* > 0.05).

The baseline and (adjusted) follow-up values for the main study outcomes, and between-group comparisons, are presented in Figure 2. The primary analyses revealed no significant between-group differences between changes in PWV (primary aim) at week 12 (median difference −0.034, 95% CI −0.42 to 0.36 m/s; *p* = 0.860; Table 3), and these results were consistent across sensitivity analyses (Table 4 and Appendix A). Regarding the secondary study aim, there were no between-group differences in the changes in hsCRP, TNF-α, IL-6, and MPO at week 12 (all *p* > 0.05; Table 3). Appendix A shows a lack of between-group differences in the change from baseline to week 12 for traditional CVD risk factors such as blood pressure, insulin resistance, or BMI. In comparison to the control group, the exercise group experienced a significant increase in cardiorespiratory fitness (median difference 2.26 min, 95% CI 0.98 to 3.55; *p* = 0.001; Table 3). Overall, sensitivity analyses corroborated these results (Table 4 and Appendix A). 

## 4. Discussion

The main findings of this study suggest that 12 weeks of progressive treadmill aerobic exercise following the ACSM guidelines, performed between 40% and 75% of the HRR, increases cardiorespiratory fitness without exacerbating arterial stiffness, inflammation, or oxidative stress in women with SLE with mild/inactive disease, in comparison to a control group of patients with SLE that received recommendations for a healthy lifestyle. Future clinical trials with larger sample sizes are needed to enhance our understanding on how different durations, types, volumes, and exercise intensities might affect vascular health and inflammation in this population.

To the best of our knowledge, this is the first study assessing the extent to which a progressive aerobic exercise intervention of moderate to vigorous intensity, following the ACSM guidelines for aerobic exercise, might influence arterial stiffness in women with SLE. Barnes et al. [40] cross-sectionally observed that participants (mainly women) with SLE who reported exercising regularly had lower central arterial stiffness than sedentary SLE subjects and similar to that of healthy individuals. By contrast, Morillas-de-Laguno et al. observed a lack of association of accelerometer-based physical activity with PWV in women with SLE [19]. Our results did not evidence any significant change in arterial stiffness following 12 weeks of aerobic exercise, which might have different explanations. Firstly, it is possible that aerobic exercise does not actually influence arterial stiffness, although this seems unlikely in light of recent evidence suggesting that this type of training reduces arterial stiffness [21,41,42]. As it has been suggested that the volume, duration, and intensity of exercise might play a relevant role in its effects on arterial stiffness [43], it seems plausible that 12 weeks of intervention might not have been enough time to modify the elasticity of the arteries. Therefore, longer interventions are warranted to unravel the effects of exercise on arterial stiffness in SLE. Secondly, the median baseline PWV in the present study (i.e., 6.3 m/s) was far from the levels considered harmful (i.e., 10 m/s) [44]. According to Ashor et al., the effects of exercise might be more pronounced in women with SLE with higher baseline PWV [21] (e.g., with older age or at higher CVD risk).

Barnes et al. [40] observed higher hsCRP and TNF-α levels in sedentary versus physically active patients with SLE. Perandini et al. observed that a single bout (i.e., 30 min) of aerobic exercise (i.e., either at 50% or 70% of VO_2peak_) did not acutely increase inflammation in women with either active or inactive SLE [22], and that a 12-week aerobic exercise program reduced resting state inflammation. However, the sample size by Perandini et al. was rather low (*n* = 8 SLE patients), the intervention was not described with complete detail to allow replication, and their results were derived from intra-group (instead of between-group) comparisons, which might be misleading [45]. In line with our results, Timoteo et al. did not observe changes in IL-6 or TNF-α following combined (i.e., flexibility, resistance, and aerobic) exercise in a rather small study (*n* = 14). The results of the present study, in sum to prior evidence, suggest that aerobic exercise training does not exacerbate inflammatory markers in women with SLE with mild/inactive disease. As exercise has emerged in recent years as a highly beneficial intervention for the cardiovascular prevention of patients with SLE [46], future clinical trials should elucidate whether diverse exercise configurations might reduce inflammation in this population. It must be noted that, in animal models, aerobic exercise has been shown to protect against the cardiometabolic disturbances induced by the chronic use of corticosteroids, such as hyperglycemia, dyslipidemia, liver steatosis, and muscular hypotrophy [47]. In this line, a positive influence of exercise on the effect of corticosteroid use on CVD risk could similarly be expected in humans, although this far exceeds the purposes of the present study and is a matter of further investigation.

Oxidative stress has been implicated in the pathogenesis of CVD in both the general population and in patients with SLE [48]. Myeloperoxidase is a biomarker of oxidative stress that predicts CVD [49]. Although previous research has described an antioxidant effect of exercise [50], our results indicate no significant reduction of oxidative stress in the exercise group compared to the usual care group, although there seemed to be a non-significant trend towards reduction in the exercise group that might deserve further research. As the sample size was not calculated based on this outcome, increasing the number of participants and combining different intensity protocols would provide valuable information regarding the association of stress oxidative and exercise in SLE.

It must be noted that the exercise intensity gradually increased throughout the intervention period, and that participants’ affective responses after each exercise session was better than before the session (Appendix A). Importantly, the exercise program significantly increased cardiorespiratory fitness, as assessed by the time to achieve 85% of estimated HR_max_ in the Bruce test [37]. As there are no SLE-specific validated formulas to estimate VO_2max_, it is difficult to assess the effects of the program on VO_2max_. However, as a reference, if we used the formula for estimating VO_2max_ in a healthy adult population described by Bruce et al. [37], the 2.26 minutes increase in the exercise group would translate into approximately 7.6 mL/kg × min^-1^ (95% CI 3.3 to 11.9; *p* = 0.001), which means >2 metabolic equivalents (METs). This increase in fitness is clinically relevant since a 1-MET increase is associated with a 13% to 15% reduction in CV and all-cause mortality [51], and with 10% to 30% lower adverse cardiovascular event rates [25]. Moreover, higher cardiorespiratory fitness has been shown to be related to a lower CV risk in patients with rheumatoid arthritis [52] and with lower age-related arterial stiffness in SLE [20], indicating that future studies should address the impact of long-term changes in cardiorespiratory fitness on CV health in SLE and other rheumatic populations.

This study has limitations. First, the sample size was relatively small and future studies with larger samples are needed. Second, we undertook a non-randomized design, which limits the confidence about baseline groups comparability and, despite statistical adjustment, residual confounding cannot be discarded. Future studies could overcome this limitation by performing a more pragmatic trial with home-based exercise and periodic visits to the clinic for outcome assessment. Third, only women with mild/inactive disease were included. Consequently, the results are not generalizable to men or even women with medium-high disease activity. Finally, despite the importance of comparative studies to better understand the early onset of atherosclerosis in rheumatic diseases [53], the present study failed to gather data from healthy individuals. Therefore, future studies should assess the comparative effectiveness of exercise in healthy people and patients with SLE. The study also has strengths that need to be highlighted. First, the adherence to the exercise intervention was very high, and the effects of the aerobic program on cardiorespiratory fitness underline that the program was effective. The intervention is described following the CERT guidelines [28], so as to enhance transparency and replicability, something that has been traditionally lacking in exercise-based clinical trials [54]. This allows any physician, physiotherapist, or exercise professional to use this exercise program in clinical practice or any other settings, thus increasing the transfer/dissemination of scientific knowledge to society.

In conclusion, the results of this study suggest that 12 weeks of progressive treadmill aerobic exercise following the ACSM guidelines increases cardiorespiratory fitness without exacerbating arterial stiffness, inflammation, or oxidative stress in women with SLE with mild/inactive disease, in comparison to a control group that received recommendations for a healthy lifestyle.

Future clinical trials with larger sample sizes are needed to enhance our understanding on how different durations, types, volumes, and intensities of exercise might affect vascular health and inflammation in this population. In particular, it seems interesting to assess the extent to which resistance training alone or in combination with aerobic training [42] might influence these outcomes in patients with SLE.

## Figures and Tables

**Figure 1 jcm-07-00477-f001:**
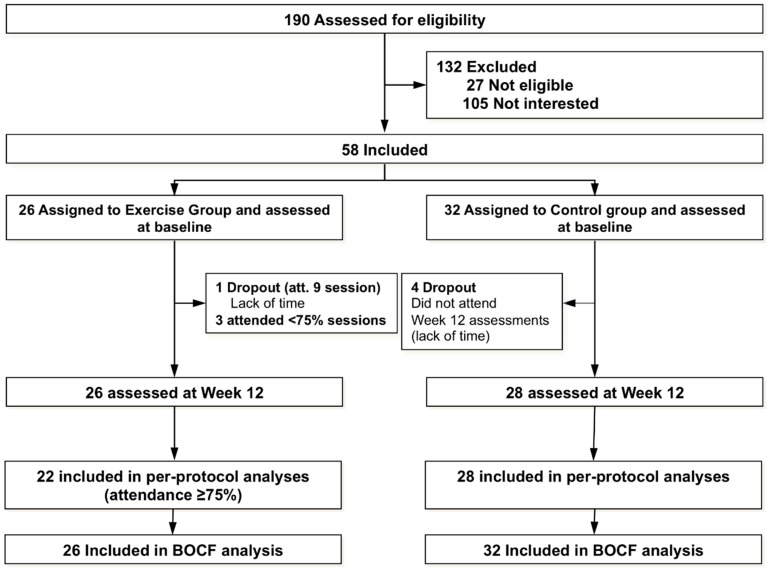
Flow chart of the study participants throughout the study.

**Figure 2 jcm-07-00477-f002:**
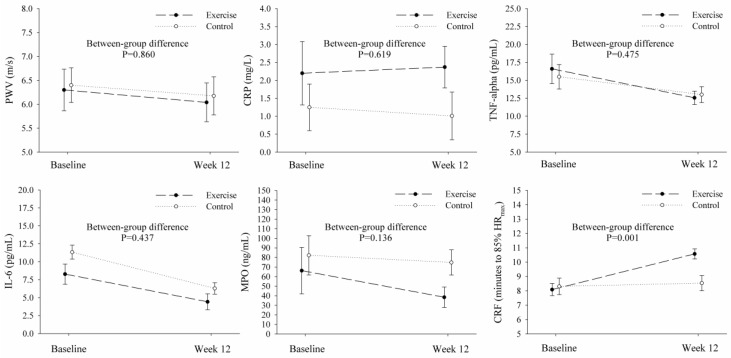
Graphical representation of the effects of the 12-week progressive aerobic exercise intervention. Results derived from the primary analyses (i.e., per-protocol with a minimum attendance of 75% for participants in the exercise group to be included). PWV, pulse wave velocity; hsCRP, high sensitivity C-reactive protein; TNF-α, tumor necrosis factor alpha; IL-6, interleukin-6; MPO, myeloperoxidase; CRF, cardiorespiratory fitness; HR_max_, maximal heart rate.

**Table 1 jcm-07-00477-t001:** Summary of the prescribed training program.

Month	Week	Weekly MVPA (min)	Session (No.)	Training Type	Total Session Time (min)	Estimated Session Time at Target Intensity (min)	Intensity (% HRR)		Series/Workout
1	1	90	1	Continuous	55	≈40	35–45		7,5’ Warm-up + 15’ 35–45% + 3’ rec + 10’ 35–45% + 3’ rec + 10’ 35–45% + 7,5’ Cool down
			2	Continuous	65	≈50	35–45		7,5’ Warm-up + 2 × (15’ 35–45%/3’ rec) + 15’ 35–45% + 7,5’ Cool down
	2	105	3	Continuous	65	≈50	40–50		7,5’ Warm-up + 20’ 40–50% + 3’ rec + 15’ 40–50% + 3’ rec + 10’ 40–50% + 7,5’ Cool down
			4	Continuous	75	≈60	40–50		7,5’ Warm-up + 2 × (20’ 40–50%/3’ rec) + 15’ 40–50% + 7,5’ Cool down
	3	130	5	Continuous	75	≈60	45–55		7,5’ Warm-up + 25’ 45–55% + 3’ rec + 20’ 45–55% + 3’ rec + 10’ 45–55% + 7,5’ Cool down
			6	Continuous	85	≈70	45–55		7,5’ Warm-up + 2 × (25’ 45–55%/3’ rec) + 15’ 45–55% + 7,5’ Cool down
	4	145	7	Continuous	85	≈70	45–55		7,5’ Warm-up + 30’ 45–55% + 3’ rec + 25’ 45–55% + 3’ rec + 10’ 45–55% + 7,5’ Cool down
			8	Continuous	90	≈75	45–55		7,5’ Warm-up + 1 × (40’ 45–55%/3’ rec) + 30’ 45–55% + 7,5’ Cool down
							Interval lower bound	Interval higher bound	
2	5	150	9	Continuous	90	≈75	50–60		7,5’ Warm-up + 30’ 50–60% + 2.5’ rec + 25’ 50–60% + 2.5’ rec + 15’ 50–60% + 7,5’ Cool down
			10	Continuous	90	≈75	50–60		7,5’ Warm-up + 1 × (40’ 50–60%/5’ rec) + 30’ 50–60% + 7,5’ Cool down
	6	150	11	Continuous	90	≈75	55–60		7,5’ Warm-up + 1 × (40’ 55–60%/5’ rec) + 30’ 55–60% + 7,5’ Cool down
			12	Interval	90	≈75	50–55	60–65	7,5’ Warm-up + 1 × (25’ 50–55%+10’ 60–65%+5’ rec) + 1 × (25’ 50–55%+10’ 60–65%) + 7,5’ Cool down
	7	150	13	Continuous	90	≈75	57.5–62.5		7,5’ Warm-up + 1 × (45’ 57.7–62.5%/5’ rec) + 25’ 57.5–62.5% + 7,5’ Cool down
			14	Interval	90	≈75	52.5–57.5	60–65	7,5’ Warm-up + 1 × (20’ 52.5–57.5%+15’ 60–65%+5’ rec) + 1 × (20’ 52.5–57.5%+15’ 60–65%) + 7,5’ Cool down
	8	150	15	Continuous	90	≈75	55–60		7,5’ Warm-up + 1 × (40’ 55–60%/5’ rec) + 30’ 55–60% + 7,5’ Cool down
			16	Interval	90	≈75	50–55	60–65	7,5’ Warm-up + 1 × (30’ 55–60%+5’ 60–65%+5’ rec) + 1 × (30’ 55–60%+5’ 60–65%) + 7,5’ Cool down
							Interval lower bound	Interval higher bound	
3	9	150	17	Interval	90	≈75	55–60	65–70	7,5’ Warm-up + 2 × (15’ 55–60%+3’ 65–70%) + 5’ rec + 15’ 55–60% + 3’ 65–70% + 12’ 55–60% + 3’ 65–70% + 7,5’ Cool down
			18	Interval	90	≈75	55–60	65–70	7,5’ Warm-up + 2 × (15’ 55–60%+3’ 65–70%) + 5’ rec + 15’ 55–60% + 3’ 65–70% + 12’ 55–60% + 3’ 65–70% + 7,5’ Cool down
	10	150	19	Interval	90	≈75	57.5–62.5	65–70	7,5’ Warm-up + 3 × (11’ 57.5–62.5% + 3’ 65–70%) + 5’ rec + 2 × (11’ 57.5–62.5% + 3’ 65–70%) + 7,5’ Cool down
			20	Interval	90	≈75	57.5–62.5	70–75	7,5’ Warm-up + 3 × (11’ 57.5–62.5% + 3’ 70–75%) + 5’ rec + 2 × (11’ 57.5–62.5% + 3’ 70–75%) + 7,5’ Cool down
	11	150	21	Interval	90	≈75	60–65	70–75	7,5’ Warm-up + 2 × (15’ 60–65%+3’ 70–75%) + 5’ rec + 15’ 60–65% + 3’ 70–75% + 12’ 60–65% + 3’ 70–75% + 7,5’ Cool down
			22	Interval	90	≈75	60–65	70–75	7,5’ Warm-up + 2 × (15’ 60–65%+3’ 70–75%) + 5’ rec + 15’ 60–65% + 3’ 70–75% + 12’ 60–65% + 3’ 70–75% + 7,5’ Cool down
	12	150	23	Interval	90	≈75	60–65	70–75	7,5’ Warm-up + 5 × (5’ 60–65%+3’ 70–75%) + 5’ rec + 2 × (11’ 60–65%+3’ 70–75%) + 7,5’ Cool down
			24	Interval	90	≈75	60–65	70–75	7,5’ Warm-up + 5 × (5’ 60–65%+3’ 70–75%) + 5’ rec + 3 × (6’ 60–65%+4’ 70–75%) + 7,5’ Cool down

MVPA, moderate-to-vigorous physical activity; HRR, heart rate reserve; Rec, recovery.

**Table 2 jcm-07-00477-t002:** Baseline descriptive characteristics of the study participants.

	All (*n* = 58)	Exercise (*n* = 26)	Control (*n* = 32)	*p*
	Mean (SD)	Mean (SD)	Mean (SD)	
Age, years	44.0 (13.9)	43.0 (15.1)	44.8 (13.1)	0.618
Marital status (Single/Married/Divorced; %)	44.8/50.0/5.2	53.8/42.3/3.9	37.5/56.3/6.2	0.455
Educational level (No studies/Primary/Secondary/University; %)	3.4/36.2/22.4/37.9	0/38.5/26.9/34.6	6.3/34.4/18.7/40.6	0.521
Occupational status (working/housewife/Not working; %)	41.4/24.1/34.5	42.3/19.2/38.5	40.6/28.1/31.3	0.706
BMI, kg/m^2^	25.2 (4.7)	25.9 (3.4)	24.7 (5.6)	0.336
SBP, mm/Hg	117.7 (10.3)	116.8 (10.0)	118.4 (10.6)	0.567
DBP, mm/Hg	75.5 (9.5)	75.6 (8.8)	75.4 (10.1)	0.937
MBP, mm/Hg	94.8 (8.8)	94.5 (8.3)	95.0 (9.3)	0.821
RHR, bpm	81.6 (11.8)	86.9 (11.0)	77.3 (10.7)	0.002
Insulin, mg/dL	7.6 (3.5)	7.7 (4.2)	7.5 (2.8)	0.809
BP lowering drugs (%)	15.5	7.7	21.9	0.138
Smoke (%)	25.9	15.4	34.4	0.166
Alcohol (yes/no; %)	5.2	7.7	3.2	0.435
Menopause (%)	39.7	38.5	40.6	0.867
History of CVD (%)	12.1	15.4	9.4	0.485
Dislipemia	17.2	19.2	15.6	0.718
Statins	17.2	23.1	12.5	0.289
Hydroxicloroquine (%)	89.7	96.1	84.4	0.143
Dosis of Hydroxicloroquine, mg/day	189.4 (115.8)	187.4 (94.1)	191.1 (132.3)	0.905
Immunosupressants (%)	44.8	46.1	43.7	0.874
Current corticosteroid intake (mg/day)	3.9 (5.1)	4.1 (6.1)	3.7 (4.1)	0.760
Cumulative corticosteroid intake (mg)	2947	2696	3164	0.511
Disease duration, years	15.4 (10.5)	14.5 (10.4)	16.1 (10.6)	0.570
Total PA, min/week	90.9 (92.2)	96.8 (97.9)	86.3 (88.8)	0.646
SLEDAI	0.22 (0.90)	0.04 (0.20)	0.38 (1.18)	0.158
SDI	0.47 (1.11)	0.19 (0.63)	0.69 (1.35)	0.092
Pulse wave velocity, m/s (median, IQR)	6.3 (5.3–7.4)	6.3 (5.1–7.8)	6.4 (5.5–7.3)	0.696
hsCRP, mg/L (median, IQR)	1.46 (0.84–4.37)	2.18 (1.2–4.37)	1.21 (0.73–4.35)	0.161
TNF-alpha, pg/mL (median, IQR)	15.95 (12.02–21.9)	16.48 (12.48–21.46)	15.18 (11.4–21.93)	0.487
IL-6, pg/mL (median, IQR)	10.33 (7.07–13.67)	8.18 (5.82–11.89)	11.27 (9.21–14.14)	0.026
MPO, ng/mL (median, IQR)	71.8 (42.8–127.1)	60.15 (40.08–113.54)	79.00 (56.65–161.1)	0.241
Cardiorespiratory fitness (Bruce test, min)	8.2 (2.8)	8.1 (2.2)	8.3 (3.2)	0.763

Values are the mean (standard deviation; SD), unless otherwise indicated. BMI, body mass index; SBP, systolic blood pressure; DBP, distolic blood pressure; MBP, mean blood pressure; CVD, cardiovascular disease; RHR, resting heart rate; PA, physical activity; SLEDAI, systemic lupus erythematosus disease activity index; SDI, systemic damage index; PWV, pulse wave velocity; hsCRP, high sensitivity C-reactive protein; TNF-α, tumor necrosis factor alpha; IL-6, interleukin-6; MPO, myeloperoxidase.

**Table 3 jcm-07-00477-t003:** Per-protocol (primary) analyses assessing the effects of 12-week progressive aerobic exercise on arterial stiffness, inflammation, oxidative stress, and cardiorespiratory fitness in women with systemic lupus erythematosus (participants in the exercise group were included if attendance was ≥75%).

Change from Baseline at Week 12	Intervention		Mean Difference (95%CI)	*p*
	Exercise (*n* = 22)	Control (*n* = 28)		
	Median (SE)	Median (SE)		
PWV, m/s	−0.26 (0.14)	−0.22 (0.13)	−0.034 (−0.42 to 0.36)	0.860
hsCRP, mg/L	0.17 (0.59)	−0.24 (0.55)	0.411 (−1.25 to 2.07)	0.619
TNF-α, pg/mL	−4.04 (1.53)	−2.49 (1.41)	−1.55 (−5.87 to 2.78)	0.475
IL-6, pg/mL	−3.86 (1.04)	−5.05 (0.99)	1.19 (−1.86 to 4.25)	0.437
MPO, ng/mL	−27.84 (9.61)	−7.49 (9.02)	−20.35 (−47.38 to 6.68)	0.136
Cardiorespiratory fitness (Bruce), min	2.49 (0.44)	0.22 (0.41)	2.26 (0.98 to 3.55)	0.001

The analyses were adjusted for baseline values, resting heart rate, and changes in self-reported physical activity during the study period. SE, standard error; CI, confidence interval; PWV, pulse wave velocity; hsCRP, high sensitivity C-reactive protein; TNF-α, tumor necrosis factor alpha; IL-6, interleukin-6; MPO, myeloperoxidase.

**Table 4 jcm-07-00477-t004:** Sensitivity analyses: Baseline-observation carried forward imputation assessing the effects of 12-week progressive aerobic exercise on arterial stiffness, inflammation, oxidative stress, and cardiorespiratory fitness in women with systemic lupus erythematosus.

Change from Baseline at Week 12	Intervention		Mean Difference (95%CI)	*p*
	Exercise (*n* = 26)	Control (*n* = 32)		
	Median (SE)	Median (SE)		
PWV, m/s	−0.25 (0.12)	−0.09 (0.12)	−0.16 (−0.51 to 0.19)	0.365
hsCRP, mg/L	−0.04 (0.36)	−0.09 (0.34)	0.06 (−0.95 to 1.06)	0.913
TNF-α, pg/mL	−4.14 (1.85)	−3.88 (1.76)	−0.26 (−5.51 to 5.00)	0.923
IL-6, pg/mL	−3.81 (1.34)	−5.02 (1.31)	1.22 (−2.79 to 5.22)	0.545
MPO, ng/mL	−17.11 (11.01)	−10.52 (10.84)	−6.58 (−37.97 to 24.80)	0.676
Cardiorespiratory fitness (Bruce), min	2.71 (0.38)	0.29 (0.36)	2.42 (1.31 to 3.52)	<0.001

The analyses were adjusted for baseline values, resting heart rate, and changes in self-reported physical activity during the study period. SE, standard error; CI, confidence interval; PWV, pulse wave velocity; hsCRP, high sensitivity C-reactive protein; TNF-α, tumor necrosis factor alpha; IL-6, interleukin-6; MPO, myeloperoxidase.

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
