# Peer review of "Effects of 12-week Aerobic Exercise on Arterial Stiffness, Inflammation, and Cardiorespiratory Fitness in Women with Systemic LUPUS Erythematosus: Non-Randomized Controlled Trial"

_jcm, 2018, doi:10.3390/jcm7120477_

Reviewer 1 Report
The manuscript reads. I have only minor comments.
Introduction. Please justify relevance of TNF-alpha and IL-6 within the frames of the pathophysiology and SLE.
Methods. It is advisable to present reproducibility data for PWV (intra- and interobserver variability).
How many patients had cardiovascular comorbidities?
Why there are no patients treated with biologics (belimumab)?
Limitations. It would be appropriate to have disease and healthy controls under the same aerobic exercise follow-up for 12 weeks. This limitation could be discussed with analysing the following relevant item: https://www.ncbi.nlm.nih.gov/pubmed/19758114
Minor copy editing is required.
Author Response
Reviewer comment: The manuscript reads. I have only minor comments.
Authors response: Thank you for the positive evaluation of our manuscript.
Reviewer comment: Introduction. Please justify relevance of TNF-alpha and IL-6 within the frames of the pathophysiology and SLE.
Authors response: This information has been included into the text (page 3, lines 65-68): “In SLE, inflammation and tissue damage are mediated by cytokines, such as TNF-alpha and IL-6, released by recruited inflammatory cells (macrophages, myeloid dendritic cells, pathogenetic T and B cells…) and immune complexes-induced complement activation” [1].
Reviewer comment: Methods. It is advisable to present reproducibility data for PWV (intra- and interobserver variability).
Authors response: The information requested by the reviewer has been included (page 11, lines 205-207).
Reviewer comment: How many patients had cardiovascular comorbidities?
Authors response: Cardiovascular disease was an exlusion criteria in the present study.
Reviewer comment: Why there are no patients treated with biologics (belimumab)?
Authors response: Clinical and treatment stability during the previous 6 months was an inclusion criteria in this study, whereas being under biological treatment (including belimumab) in the previous 6 months was an exclusion criteria.
Reviewer comment: Limitations. It would be appropriate to have disease and healthy controls under the same aerobic exercise follow-up for 12 weeks. This limitation could be discussed with analysing the following relevant item: https://www.ncbi.nlm.nih.gov/pubmed/19758114
Authors response: The article recommended by the reviewer has been considered and included in the discusison of the limitations of our study (page 22, lines 423-426). Thank you.
Reviewer comment: Minor copy editing is required.
Authors response: The manuscript has been carefully revised and minor typos/errors have been corrected.
References
1. Zharkova, O.; Celhar, T.; Cravens, P.D.; Satterthwaite, A.B.; Fairhurst, A.M.; Davis, L.S. Pathways leading to an immunological disease: Systemic lupus erythematosus. Rheumatology (Oxford) 2017, 56, i55-i66.
Reviewer 2 Report
The authors are describing the usefulness of certain exercise for increase of cardiorespiratory fitness. I think this increase will lead to long-term exercise improving arteriosclerosis estimated by exacerbation of arterial stiffness even in patients with SLE. However, I wonder the authors just mentioned about the changes of PWV and proinflammatory cytokines to evaluate arteriosclerosis as a risk factor for CVD, because the most important risk factor in patients with SLE on CVD is consecutive corticosteroids treatment. Therefore, the authors should show or discuss about the effect of corticosteroid on CVD risk and how exercise exacerbates CVD risk by corticosteroid.
Author Response
Reviewer comment: The authors are describing the usefulness of certain exercise for increase of cardiorespiratory fitness. I think this increase will lead to long-term exercise improving arteriosclerosis estimated by exacerbation of arterial stiffness even in patients with SLE.
Authors response: Thank you for the positive evaluation of our manuscript.
Reviewer comment: However, I wonder the authors just mentioned about the changes of PWV and proinflammatory cytokines to evaluate arteriosclerosis as a risk factor for CVD, because the most important risk factor in patients with SLE on CVD is consecutive corticosteroids treatment. Therefore, the authors should show or discuss about the effect of corticosteroid on CVD risk and how exercise exacerbates CVD risk by corticosteroid.
Authors response: We agree with the reviewer that cumulative corticosteroid intake is a well-known major CVD risk factor in patients with SLE. However, this harmful effect of corticosteroids seems to be specially relevant when receiving a daily dose higher than 10 mg [1], a cumulative dose >10mg/day over 10 years and/or after long disease duration [2]. In the present study, the current corticosteroid intake was 2.9 mg/d and the cumulative corticosteroid intake was 2947 mg (mean cumulative dose of 0.5 mg). Importantly, it must be noted that we accounted for (both current and cummulative) corticosteroid intake by assessing between-group differences on these variables at baseline. As can be observed in table 2, the groups were comparable in regards to both current (P=0.760) and cumulative (P=0.511) corticosteroid intake.
In animals models, aerobic exercise has shown to protect against the cardiometabolic disturbances induced by the chronic use of corticosteroids, such as hyperglycemia, dyslipidemia, liver steatosis and muscular hypotrophy [3]. Therefore, it would be expected a positive influence of exercise on the effect of corticosteroid use on CVD risk, although this far exceeds the purposes of the present study.
This information has not been included in the manuscript, although we are open for possible inclusion should the reviewer or editor consider it necessary.
References
1. Magder, L.S.; Petri, M. Incidence of and risk factors for adverse cardiovascular events among patients with systemic lupus erythematosus. Am J Epidemiol 2012, 176, 708-719.
2. Kravvariti, E.; Konstantonis, G.; Sfikakis, P.P.; Tektonidou, M.G. Progression of subclinical atherosclerosis in systemic lupus erythematosus versus rheumatoid arthritis: The impact of low disease activity. Rheumatology (Oxford) 2018. [Ahead of print]
3. Pinheiro, C.H.; Sousa Filho, W.M.; Oliveira Neto, J.; Marinho Mde, J.; Motta Neto, R.; Smith, M.M.; Silva, C.A. Exercise prevents cardiometabolic alterations induced by chronic use of glucocorticoids. Arq Bras Cardiol 2009, 93, 400-408, 392-400.